# Conceptualizing a Product with the Food-Related Lifestyle Instrument

**DOI:** 10.3390/foods11223549

**Published:** 2022-11-08

**Authors:** Oxana Lazo, Luis Guerrero, Sergio Erick García-Barrón

**Affiliations:** 1CIBA, Instituto Politécnico Nacional, Research Center for Applied Biotechnology, Carretera Estatal Santa Inés Texcuexcomac Km 1.5, Tepetitla 90700, Tlaxcala, Mexico; 2Food Quality and Tecnhology, Institute of Agrifood Research and Technology (IRTA), Finca Camps i Armet s/n, Monells, E-17121 Girona, Spain; 3CIATEJ, Research and Assistance Center in Technology and Design of the State of Jalisco A. C., Av. De los Normalistas #800, Colinas de La Normal, Guadalajara 44270, Jalisco, Mexico

**Keywords:** product conceptualization, food-related lifestyle constructs, mezcal, consumer behavior

## Abstract

Product perception is important for consumers’ acceptance, especially when it is associated with a geographical location. Consumers’ food-related lifestyles (FRLs) have been used to better identify the role that beverages have in people lives. The present study was conducted to understand the conceptualization of mezcal according to consumers’ FRLs. Four hundred mezcal consumers were surveyed in Mexico. Participants were asked to describe their experience with the product and consumption habits, to evaluate ten different FRL constructs, and to assess mezcal conceptualization using a check-all-that-apply test. A hierarchical cluster analysis was carried out on the composite variables of the evaluated constructs and their objective knowledge score to define segments. To visualize the relationships among FRL constructs and the terms used to describe mezcal, a multiple factorial analysis was carried out. The results showed four different mezcal clusters. The social and involved segment described the beverage with elements of traditional and food-related activities. The price–quality fixed segment was mainly associated with the product to handcraft process. Uninvolved consumers were not linked to specific terms and uninformed and unaware consumers were novice participants with mainly negative product connotations. Therefore, is important to consider consumers’ FRLs to have a better understanding of product conceptualization.

## 1. Introduction

Food product representation can be defined through the group of characteristics that consumers use to describe a product and it is very relevant since the image created can influence consumers’ choice and consumption [1]. Therefore, it is important to understand and identify how the relative meaning of a product is built, and what is the general conceptualization that consumers have towards that product [2]. Such a set of elements may be the reflection of consumers’ practices [3,4], experiences derived from consumption [5], as well as consumption habits and consumers’ lifestyles [6,7].

Lifestyles are defined as patterns in which people live and spend time and money [8]. Lifestyles are a function of consumers’ motivations and prior learning, social class, demographics, among other variables. In this vein, lifestyles have also been regarded as mental constructs that explains people’s behaviors [9]. A FRL instrument has been created and validated for measuring the meaning of food in people’s lives. According to Grunert [10], FRL is composed of different elements:(a)Quality aspects such as the price–quality relationship and novelty;(b)Purchasing motives such as self-fulfillment in food and social relationships;(c)Consumption situations such as social events;(d)Ways of shopping, importance of product information, and price criterion.

Furthermore, everyday observations suggest that the role of food in life differs among people [11]. Therefore, people differ in the degree of their involvement with food [12], and they also differ in the reasons for the degree of involvement that they have with food as part of their lifestyle [11]. Thus, food involvement can be defined as an individual’s perception of the relevance of a product based on needs, values, and interests [13]. In addition, FRL also considers knowledge as a construct that provides behavioral routines to act upon the mental representation of a product [7]. Thus, the level of knowledge that a consumer has about a product also influences their attitude, purchase intention, and expectations [14,15]. As a construct, the level of consumer knowledge can be evaluated based on the objective and subjective knowledge they have about a product of interest [16]. Objective knowledge refers to the information stored in memory and how it is organized, which represents what a consumer really knows about the product in question. In contrast, subjective knowledge is made up of a consumer’s personal perceptions of how much knowledge they think they have about a product [17,18,19].

All these consumer psychographic factors (objective and subjective knowledge, novelty, product information, price–quality relationship, price criterion, product involvement, self-fulfillment, social events, and social relationships) can be decisive when shaping relationships between concepts and products, especially when they are linked to a geographical location [20,21].

Several studies have employed the FRL instrument, mostly with the objective of deriving segments of consumers that differ in the role that food plays in their lives [9]. This technique has been used to explore the attitudes and behavior patterns of consumers towards a product [22]; additionally, it has been used to analyze the degree of consumers’ product involvement [11]. Some examples of the instrument’s applications are segmentation of consumers according to their wine-related lifestyles [23] and convenience food-related lifestyles for vegetables [24]. All these studies have been used as predictors of food-related behavior, but they have not assessed the relationship of this behavior with a product’s description (concept). Therefore, to the best of our knowledge, the influences of the aforementioned FRL constructs on product conceptualization have not been explored. Since the FRL instrument allows a consumer’s behavior towards a food product to be understood, in this study, we selected it as a methodological approach to assess its relation to word selection when conceptualizing a product.

Given this context, consumers’ lifestyles influence the conceptualization of a typical alcoholic Mexican beverage, namely mezcal. Mezcal is a traditional beverage with cultural and economic importance. Its consumption has grown significantly in recent years [16], especially since it acquired protected designation of origin status in different Mexican regions, which provides a sensory identity linked to production sites. Since 2015, Mezcal production has increased up to 147%, due to its growing number of producers, its diversity, and recent diffusion. Consequently, this product has been placed in international markets, including the European market [25]. A previous work, studied the mezcal concept by observing the words associated with mezcal per region [26]. However, this study did not address the connection of these words to an individual’s characteristics. Therefore, it would be important to link the use of these terms to consumers’ FRL features, to gain a better understanding of their selection. There are different approaches to study the representation of a concept. One approach is the use of check-all-that-apply questions, since it is an easy approach for consumers to perform [27,28]. This methodology basically consists of asking consumers to select all the terms from a list that apply to describe a focal product [29].

To sum up, the aim of this study was to assess how consumers conceptualize a Mexican beverage, i.e., mezcal, using CATA words, and to link the descriptions obtained to the evaluated constructs. Therefore, three hypotheses were proposed. The first hypothesis is that consumers’ food-related lifestyles vary depending on the city. The second hypothesis is that mezcal conceptualization in each city is different due to an effect on term selection. Finally, the third hypothesis is that there is a relationship between the terms selected for mezcal conceptualization and consumers’ food-related lifestyles (constructs).

## 2. Materials and Methods

### 2.1. Participants

Four hundred consumers from four different Mexican cities participated in this study. The selected cities were Mexico City, because it represents the largest mezcal consumption market in Mexico [16]; Guadalajara, which is in western Mexico where Mezcal consumption is incipient; the southern city of Oaxaca, because it represents the area with the highest production of mezcal in the country; and finally, Puebla, as a mezcal production zone that recently obtained the protected designation of origin seal [16]. In each city, 100 consumers were surveyed face-to-face using convenience sampling. Convenience sampling is a technique frequently used in social and marketing research, because participants are selected by accessibility and proximity to the researcher [30]. The main inclusion criteria were: an interest in participating; over 18 years old; and a mezcal consumer on, at least, a monthly basis.

### 2.2. Instrument

The questionnaire consisted of three sections: (a) sociodemographic characteristics of consumers, experience with the product, and consumption habits (Table 1), (b) evaluation of ten different constructs (Table 2); (c) assessment of Mezcal conceptualization using a check-all-that-applies (CATA) test.

#### 2.2.1. Sociodemographics, Experience, and Consumption Habits

Sociodemographic variables (gender, age, level of education, and occupation), experience with the product, and consumption habits were collected. Five age groups were considered: from 18 to 24 years old, from 25 to 34 years old, from 35 to 44 years old, from 45 to 54 years old, and 54 years old and older. Two levels of education were sampled: undergraduate and postgraduate. Four categories of occupation were considered: employed, self-employed, student, and unemployed. Five categories of experience with the product were taken into account: less than a year, from one to three years, from three to five years, from five to seven years, and more than seven years. Consumption habits addressed the frequency with which mezcal was consumed including: every day, every third day, once a week, every 15 days, and once a month (Table 1).

#### 2.2.2. Constructs Evaluation

Ten different constructs were evaluated (Table 2): food involvement [11], objective knowledge (own elaboration), novelty, price criterion, price–quality relationship, product information, self-fulfillment, social events, social relationships [10], and subjective knowledge [19]. All constructs, except objective knowledge, were assessed using a seven-point Likert scale, from 1 “totally disagree” to 7 “totally agree”. Objective knowledge was evaluated by means of a 15 multiple choice questionnaire formulated by professionals of mezcal industries in collaboration with the authors of this work (Appendix A). Items in the objective knowledge questionnaire included aspects related to the composition of the product, its production and processing, quality marks, sensory properties, as well as traditions and cultural issues associated with its consumption. Although the questions were formulated by experts and checked and adapted by the authors of the study to ensure that they could be understood and answered by an average consumer of the product, a pretest of all questions was carried out with a sample of 10 regular consumers of each product. This pretest allowed us to verify that the selected items were not excessively technical or complex.

#### 2.2.3. CATA Questionnaire

To evaluate the words that conceptualize mezcal, a CATA test was used. A CATA test is a technique that has been used to characterize different type of foods [31,32]. Following the recommendations of Ares and Jaeger [28], five different order designs of CATA were developed to avoid bias (on term order effect) for the evaluated concepts. Participants were provided with a list of terms and had to indicate which terms were appropriate to define the concept of mezcal. The terms considered were agave, alcohol, blue agave, bottle, burning, cheap, commercial, culture, drink, drunk, drunkenness, food, handcraft, hangover, industrialized, industry, maguey, Mariachi, Mexican, nice, orange, paloma cocktail, party, smoked, soft, state, strong, tradition, and worm. Additionally, the option “other” was given if participants considered it necessary. The selected words in the CATA questionnaire were previously obtained by García-Barrón et al. [26], who observed that consumers associated them with mezcal.

### 2.3. Data Analysis

The construct’s unidimensionality was evaluated using a factor analysis (principal components extraction method). Only those items that were significant in the first component, with a factor loading higher than 0.50, were considered for further analysis (the rest were removed). Additionally, to check a construct’s reliability and internal consistency, a Cronbach’s alpha coefficient was computed. Then, composite variables were obtained by calculating the mean values of all items included in each construct, except for objective knowledge, where the total number of correct answers were counted (between 0 and 15). In addition, a two-way ANOVA was carried out on the evaluated quantitative constructs to assess differences among cites. Cities were considered as fixed factors, and consumers as random factors. When significant differences were observed (*p* < 0.05), a post hoc test for multiple comparisons of mean values was performed through Tukey’s HSD test.

CATA data were analyzed considering the frequency of use of the selected terms to describe the concept of the beverage (number of consumers who selected the word). To estimate the presence of significant differences (*p* < 0.05) in the use of each of the terms among the cities, a K proportions test (Marascuilo procedure) was carried out.

To identify consumer segments with similar psychographic characteristics, a hierarchical cluster analysis (Ward method and Euclidian distance) was carried out on the composite variables of the nine constructs evaluated, and the final objective knowledge score. The sociodemographic characteristics of participants and their corresponding experience and consumption habits were used to characterize the different clusters obtained. The number of segments retained was decided based on the dendrogram, taking into account the homogeneity intra and inter segments and the principle of parsimony [33]. The validation of the identified segments was carried out through a discriminant analysis to assess the percentage of individuals correctly classified in their respective cluster. Subsequently, a two-way ANOVA was carried out on the mean values of the ten constructs including the segment and the participants as fixed and random effects, respectively. Furthermore, to determine significant differences among segments a post hoc Tukey’s test (*p* < 0.05) was performed. Additionally, the frequencies of the terms selected by the members of each segment to conceptualize mezcal were analyzed. To establish the presence of significant differences among segments, a K proportion analysis using the Marascuilo procedure (*p* < 0.05) was performed on the sociodemographic characteristics, consumption habits, experience with the product, as well as on the frequencies of the selected terms in the CATA test. Finally, to visualize the relationship between the evaluated constructs and the terms used to describe mezcal according to the segments, a multiple factorial analysis (mixed version) was carried out. A multiple factor analysis (MFA) is a multivariate statistical tool that that can analyze multiple data tables, which can be quantitative, qualitative, or combined data. MFA was realized on the frequencies of each term used to describe the mezcal concept and the mean values of all the assessed constructs.

To acquire a quantitative measure of proximity between matrices (selected terms and constructs), the regression vector (RV) coefficient was calculated. RV values can range between 0 and 1, with 1 indicating the highest similarity between configurations obtained between the two matrices [34]. All the analyses were carried out by means of the XLSTAT 2019 software (Addinsoft, Paris, France).

## 3. Results

### 3.1. Constructs Validation

Constructs’ reliability was confirmed (for the nine quantitative constructs) with Cronbach alpha values of 0.86 for product involvement, 0.76 for product information, 0.58 for price as criterion, 0.82 for quality–price relationship, 0.87 for novelty, 0.82 for social events, 0.64 for self-fulfillment, 0.65 for social relationships, and 0.73 for subjective knowledge. According to George and Mallery [35], Cronbach’s Alpha values equal to or higher than 0.5 can be accepted to confirm construct reliability. The factorial loadings obtained were also significant, ranging from 0.56 to 0.99, thus, confirming the constructs’ unidimensionality (Table 2).

### 3.2. Construct Assessment among the Cities

After construct validation, some significant differences in construct scores (mean values) were found among the cities according to the Tukey test (*p* < 0.05) (Table 3). The Novelty and Price criterion were the constructs with higher mean values for mezcal consumers from all four cities (at least five out of seven points on the Likert scale). Objective knowledge, self-fulfillment, social events and subjective knowledge score values were higher for Oaxaca consumers. The price–quality relationship mean value score was slightly higher for Guadalajara consumers.

### 3.3. Concept Description of Mezcal among the Cities

Table 4 shows the proportions in which CATA terms were selected to conceptualize mezcal in each city. Similarities and differences were observed in the selected words among the cities. On the one hand, some words such as culture, handcraft, Mexican, smoked, state, strong, worm, and tradition were selected, without significant differences, by 49 percent or more participants in each city. Other words such as blue agave, industry, industrialized, and paloma cocktail were also selected equally by a smaller number of participants (10% or less) in each city. On the other hand, differences were also noticed depending on the city. One example was the use of the word “maguey” which was selected by 80% of the participants from Oaxaca, a significantly higher percentage as compared with the other cities. Another example was the word “orange” which was selected in a higher percentage by participants from Puebla as compared with the other cities as well. The term “drink” was also significantly higher for participants from Guadalajara and the word “nice” was significalty higher for consumers from Mexico City.

### 3.4. Segment Profiling

A hierarchical cluster analysis was performed on the ten scored constructs that showed four different segments of mezcal consumers (Table 5). The confusion matrix obtained through the discriminant analysis showed that 88% of consumers were correctly classified.

Table 5 shows the mean scores for the 10 assessed constructs considered for segment profiling. The results of the Tukey’s test showed that there were significant differences among segments for all 10 constructs, and especially for the food involvement construct, which had a significantly different value for each cluster.

The analysis of sociodemographic differences among segments showed no significant differences in gender composition nor in the age category. Regarding the cities, only Oaxaca exhibited a significant difference for segment characterization. Education was also a parameter for segment differentiation. In addition, differences in experience with the product and frequency of consumption were also observed among the segments (Table 6).

Cluster 1 (28.75% of the sample) was named social and involved; it was characterized by having the highest scores on the food involvement construct. It also exhibited the highest scores for social events, social relationships, and self-fulfillment constructs, being significantly different from the other three clusters (Table 5). Additionally, this segment showed the highest percentage of participants with a university degree (78.3%). It also had a higher number of participants that had been consuming the product for more than seven years (47.8%) as compared with the other clusters, and it had more participants with a higher frequency of consumption (29% every third day) (Table 6).

Cluster 2 (12.75% of the sample) was named fixed in quality; it was characterized by having the highest scores on the quality–price relationship construct (Table 5). In addition, this segment had the lowest percentage of participants from Oaxaca (5.9%), and it also showed a higher percentage of consumers that did not possess a college degree (45.1%) as compared with the other segments (Table 6).

Cluster 3 (35% of the sample) was tagged as low involved; it was characterized by having low scores on the involvement construct and it was neutral opinionated about the information, price criterion, social events, social relationships, and self-fulfillment constructs (Table 5). In addition, it had the highest percentage of consumers that consumed the product every 15 days as compared with the other clusters (Table 6).

Cluster 4 (23.5% of the sample) was named unaware and uninformed; it was characterized by having low scores on the objective knowledge and price–quality constructs and it had the lowest scores in the other eight constructs (Table 5). Along with Cluster 3, it had a higher percentage of participants that had been consuming the product between 1 and 3 years. In addition, it had the highest percentage of participants that consume the product once a month as compared with the other segments (Table 6).

### 3.5. Relationships across Constructs, Terms, and the Obtained Segments

Regarding mezcal conceptualization according to segments, the terms agave, culture, food, maguey, and tradition were the terms significantly selected by Segment 1. Participants from Segment 2 considered the terms handcraft and Mexican as part of their conceptualization, showing the highest mean values (0.92 and 0.80, respectively) on proportions of selected words as compared with the other segments. Segment 3 did not show specific terms associated with it. The terms cheap and hangover were significantly selected by Segment 4. In addition, the terms commercial, drunk, drunkenness, and strong exhibited the highest mean values among consumers from this segment (Table 7).

The first two dimensions of the MFA carried out on the mean values of assessed constructs and selected words to conceptualize mezcal per cluster are shown in Figure 1. They explain 91.43% of variation and reveal a view of mezcal conceptualization according to consumers’ food-related lifestyles. The first dimension differentiates consumers who are knowledgeable about the product from those who are not, as well as socially involved consumers from those who are not. The second dimension contrasts handcraft mezcal consumers with the industrial consumers. The obtained RV coefficient between matrices was 0.92, thus, confirming the relationship between the selected words and FRL constructs for segments.

## 4. Discussion

The present study investigated the differences in consumers’ conceptualization of a product, and how their food-related lifestyles may affect these conceptualizations. Therefore, the relationships between the FRL constructs and the selected words used to describe mezcal were assessed. First, possible influences of the cities were addressed, then, the study was oriented to establish consumer segments with similar response patterns.

### 4.1. Tested Hypothesis

The first hypothesis, i.e., consumer’s food-related lifestyles vary depending on the city, was partially confirmed by the results of the ANOVA performed on the mean values of the constructs assessed for each city (Table 3). The idea that the city influences participants FRL was significantly confirmed only for consumers from Guadalajara and Oaxaca. This premise did not apply to participants from the other cities.

Guadalajara consumers had a higher tendency (as compared with the other cities) to care about the price–quality relationship construct. A positive relationship between brand and expected quality in beverages has been previously outlined [36]. When low product familiarity exists, consumers seem to pay attention to aspects of their known brands. Thus, these consumers look for a quality–price relationship with their habitual product.

Objective knowledge was also a construct with significantly higher scores, specifically for Oaxaca consumers. Participants from this city have an ancient tradition of consumption which could explain the familiarity with the process, raw materials, methods of production, among other information gathered. According to Banović et al. [37], product familiarity has been known to have a direct relationship with product knowledge. Therefore, a greater objective knowledge may have developed with these consumers.

In addition, some constructs such as novelty and price criterion were equally important for all cities, showing mean values above the average score. Mezcal is a product fabricated across different regions in Mexico with different traditional processes, associated with a diversity of brands that give different flavors to the product. These varieties could be attractive for certain type of consumers. According to Brunsø et al. [7], consumers who pay attention to the novelty of a product can be related to people striving for stimulation. In addition, some consumers often show a tendency to seek variety, which implies trying new things as a part of a new product adventure. Moreover, consumers who look for novelty tend to be the opposite of conservative consumers, which means they constantly like to try new things [10], and which reflects the overall behavior of these participants. Price criterion was also highly considered by most participants. Price is unquestionably one of the most important marketplace cues. Price is present in all purchase situations and, at a minimum, represents to all consumers the amount of economic outlay that must be sacrificed to engage in a given purchase transaction [38]. Consumer price involvement refers to an array of psychographic dimensions that underlie consumers’ price-related marketplace behaviors [39]. Thus, it is no surprise that this construct was important for all consumers. The observed differences in construct assessment among the cities implies that there is an interaction between a city and the food-related lifestyle constructs.

The second hypothesis regarding differences in conceptualization among the cities depending on terms selection was confirmed by the K proportions test of selected words per city (Table 4). An interaction between the cities and the words used for mezcal conceptualization was noticed, since different proportions in the selection of terms were observed, which means that a city can influence the frequency which with a word is selected for the mezcal concept. There was also an agreement in some words used to conceptualize mezcal, showing a high proportion of selection by all cities. Thus, mezcal was generally perceived as a nice traditional Mexican product, made from the maguey plant with a handcraft process, linked to people’s culture.

Regarding the significant differences in the proportions for the same word among the cities, Oaxaca consumers used terms related mainly to raw material and celebrations (agave, maguey, and party). Participants from Mexico City used terms associated with hedonic characteristics and gatherings (nice and food). Consumers from Puebla selected words related to cultural modes of consumption (orange). Finally, consumers from Guadalajara used terms related to drinking experience (drink).

In general, it could be said that consumers’ geographical locations influence product description but do not influence the participants’ lifestyles. Thus, to better understand the relationships between constructs and the terms used for mezcal conceptualization, it was important to assess consumers’ mezcal perceptions according to more than their origin which was their food-related lifestyle. Therefore, conceptualization was analyzed based on selected words of consumers per established segments (based on FRL).

The third and final hypothesis regarding the interaction between the terms selected for mezcal conceptualization and consumers food-related lifestyle was validated through a multiple factor analysis and the RV coefficient obtained. The results in this work, showed that there was, indeed, a relation linking FRL constructs and terms used to define the mezcal concept (Figure 1).

### 4.2. Mezcal Conceptualization According to Segments Based on FRL

The words agave, culture, food, maguey, and tradition were significantly selected by Segment 1 (social and involved) when conceptualizing mezcal (Table 7) (Figure 1).

The connection between food and alcohol has been acknowledged for a long time; it is influenced by social, cultural, and behavior patterns [40]. In addition, the associations of beverages with specific foods and the exclusive taste these combination bring, has also been recognized as one the most important factors for alcohol consumption [41]. Therefore, the use of the terms “food” and “culture” for mezcal description could relate to social activities that link the beverage to social relationships and social events, thus, referring to these consumer habits of drinking mezcal while having a meal in social gatherings. In addition, involved consumers care about their social life and the self-fulfillment that comes with it, hence, the association of the term “food” with their mezcal conceptualization. Castellini et al. [42], stated that “being involved in food means using food to reach social affirmation”, as the relationship with food allows one to belong to a social group and to be accepted by others, which confirms this segment’s conduct.

These consumers also showed the highest percentage of participants with a college degree, the highest frequency of consumption, and had the highest number of participants who had been consuming the product for over 7 years, which indicated familiarity and product knowledge. Consumer product involvement has been positively correlated with product knowledge [43]. In this vein, maguey is the raw material used for mezcal production; in fact, the word mezcal means cooked maguey [44]. Therefore, to use this word, knowledge about the product is required. This could explain the association of the word “maguey” with mezcal for this segment of consumers. In addition, part of consumers’ objective knowledge (Appendix A) may consider the traditional process of mezcal and its cultural modes of consumption, which could explain the use of the terms “culture” and “tradition” for this product conceptualization.

Regarding Segment 2 (fixed in quality-price), some consumers are sensitive to differences in product quality. In these cases, price can assume a positive role, especially when it acts as a cue to a positive product attribute that improves its quality [39]. This segment was associated with the words “handcraft” and “Mexican”, the opposite of commercial, industrial, and industrialized (Figure 1). According to Rivaroli et al. [45], craft food products, including drinks, have attracted widespread interest among consumers during the past few years. In addition, linking higher quality to a handcraft process has been observed to increase when it comes to alcoholic beverages [2]. In general, craft products have been noted to be an indicator of better ingredients and better process techniques [46]. Therefore, the quality–price relationship for the mezcal concept could be connected to a handcraft process of a Mexican product. However, unlike Bruwer’s study [8], in this research, the consumers who paid more attention to the quality price parameter were not the most knowledgeable consumers. As a matter of fact, they exhibited a rather low score for objective knowledge and a very low score for subjective knowledge, hence, their fixation on the quality–price relationship was not triggered by product knowledge.

In Segment 3, even though these participants did select words when conceptualizing mezcal, there were no words significantly associated with this cluster (low involved), which means participants in this cluster selected terms in the same proportion as in the other clusters (Figure 1). This could probably be related to these consumers lack of opinions on the social relationships, social events, price criterion constructs, and their low involvement as well. When consumers have low involvement in food products (including drinks), they usually center their values in other domains of life, making it hard to understand the way they conceptualize this product according to the constructs assessed [11].

The words “cheap”, “drunk”, “drunkenness”, “hangover”, and “strong” were selected by Segment 4, i.e., unaware and uninformed consumers (lowest attention to product information, lowest subjective knowledge, lowest product involvement in social events) (Figure 1). The selected words in this segment seem to be related to effects of any alcoholic beverage. Therefore, the association of these terms when conceptualizing mezcal is a rather evident choice. According to Palacios and Farooq [47], a person who has limited knowledge and less experience with a product can also be defined as a novice in the product. Novice consumers that have low knowledge of products usually focus on limited characteristics, connecting with the most obvious and convenient product attributes [48]. In this vein, the term “drunkenness” has also been used by novice consumers when describing wines [5], which may indicate that the use of this term is related to low experience and low product knowledge. In a different line, novice consumers have been known to purchase their beverage at social occasions, and often use price as a determinant factor for acquisition [8]. Nevertheless, in this study, novice consumers exhibited the lowest scores on the social events construct and the price criterion construct, which means those are not their main consumption motives. In fact, these consumers showed low scores on all the FRL constructs that were assessed in this study, which means their reasons for consumption rely on different matters.

Finally, the product information, price criterion, and novelty constructs were equally important for Segments 1 and 2. The terms associated with these constructs were: soft, state, and smoked (Figure 1). Part of product information for mezcal is whether the raw material has been smoked during its process and the place of origin (state) of the product. Therefore, consumers who notice the product information could use these terms as part of their conceptualization. The term soft is related to the overall perceived alcohol intensity of a distilled product; it has been used as a sensory characteristic of different alcoholic beverages [49], and some consumers may use this characteristic as price criterion when purchasing alcoholic beverages.

### 4.3. Contributions and Practical Implications

Product conceptualization has been analyzed using different methodologies. Ares et al. [50] conceptualized ultra-processed foods with consumers using a Free listing task. Esmerino et al. [51] studied consumers’ perceptions using: pivot profile (PP), check-all-that-apply (CATA), and projective mapping (PM) to describe Greek yogurt. In 2017, the beer drinking experience was characterized using CATA phrases [52]. However, the relationship between consumers characteristics and the assessed terms in those studies was not analyzed. In addition, the FRL instrument has been previously used for consumer segmentation with the main purpose of understanding whether consumers can be profiled according to their lifestyles [9]. Yet, the instrument has not been considered for product conceptualization perse. Therefore, it was important to assess consumers’ product conceptualization according to their lifestyles.

This study addresses the gap in product conceptualization by showing one main important finding. Conceptualization is more than word selection to describe a product, it requires an understanding of a person’s characteristic to use these choices. Furthermore, there is an association between the words used to conceptualize mezcal and consumers’ lifestyles.

As part of the practical implications in this work, it is important to consider whether retailers should care if the product that they sell to consumers is perceived in the way they want or not. Some advice for mezcal retailers could be to take into consideration the mezcal segmentation in this study. The different mezcal segments should be approached individually, especially if the question of how to target consumers through mezcal conceptualization is presented.

Regarding the social and involved consumers (Segment 1), Food-involved consumers who are immersed in social events and have product knowledge, are likely to read information in a deeper way and to be driven by the product cues through label divulgence [53]. This segment could also be reached through social communication, where attractive insights can be provided to stimulate product characteristics. This could be achieved in special social events such as mezcal tastings or promotion fairs. In addition, this group could use even more detailed information on academic events [54]. Therefore, communication to this segment seems to be crucial.

Consumers who have similar behavior to the segment of “fixed in quality–price relationship” (Segment 2)*,* could less likely act on price and budget motives. Instead, marketing actions tempting their handcraft interest, could be a successful direction to increase mezcal consumption for this group. In addition, special designs such as handcraft mezcal packages offered in specialty shops could be a marketing strategy suitable for this consumers group.

Given the general low concern about product quality and absence of involvement of the “Low Uninvolved” (Segment 3) and the Unaware and unfinformed (Segment 4), these groups should not be expected to make a conscious effort to put attention on product information. These participants might not be interested enough in learning about the product nor in targeting low prices. Thus, marketing measures that provide a change in choice environment or smarter packaging, may be successful for getting the attention in these segments [55].

## 5. Conclusions

In this paper, we focus on consumers’ conceptualization of a product, in this case, mezcal. The results presented in this study provide an important vision of mezcal drinking and purchasing habits among Mexican consumers. The discoveries are interesting as they present an important understanding of the experience and variables involved in mezcal conceptualization, specifically concerning the consumers’ food-related lifestyles.

This study helps to address the gap in product conceptualization research that is related to the influence of consumer characteristics on term selection to describe a product. We propose that a product conceptualization can be better understood by taking into consideration consumers’ product knowledge, involvement, and their lifestyles.

The findings in this work indicate that product involvement is a key element when conceptualizing mezcal. Social events, social relationships, and self-fulfillment also interfere with product conceptualization as they emphasize the context in which the product is consumed. To the contrary, lack of product knowledge and product information correspond to a limited conceptualization of mezcal associated with mostly negative terms.

In addition, the existence of different segments implies differences in product conceptualization, and therefore, differences in how consumers can accept the product. The results show that even though most consumers perceive mezcal as a Mexican handcraft product, this feature is especially important for those who pursue a quality–price relationship with their product. The ”fixed in quality–price relationship mezcal drinkers” is a significant segment for mezcal marketers to target with customized handcraft products. It is important to note that, given the existence of these clusters, mezcal organizations should consider the profitability of each segment thoroughly, for instance, by targeting knowledgeable and involved mezcal consumers in social events.

On the one hand, this type of study can also help product developers to better design their products according to the degree of involvement a consumer has. On the other hand, it would be important for marketing teams to develop strategies for the uninvolved consumer.

Segmenting consumers according to how they perceive value in mezcal can also improve comprehension of consumers in other distilled markets. This segment conceptualization can improve consumers’ understanding from a lifestyle perspective; however, additional development and research work is required to improve its psychometric properties, if possible, in a cross-cultural study.

## Figures and Tables

**Figure 1 foods-11-03549-f001:**
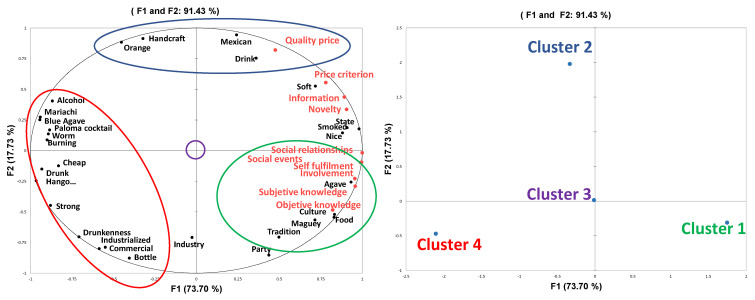
Relationship among the assessed constructs and selected terms according to clusters.

**Table 1 foods-11-03549-t001:** Sample characteristics (number of individuals).

		Guadalajara *N* = 100	Mexico City *N* = 100	Oaxaca *N* = 100	Puebla *N* = 100
Gender	Female	37	40	41	41
	Male	63	60	59	59
Age	18–24	16	10	24	12
	25–34	46	39	40	37
	35–44	29	23	27	34
	45–54	7	20	7	14
	55 or more	2	8	2	3
Education	Undergraduate	34	31	46	9
	University degree	66	69	54	91
Experience with the product	Less than a year	16	5	10	16
	From 1 to 3 years	19	17	21	26
	From 3 to 5 years	27	18	15	21
	From 5 to 7 years	20	17	16	19
	From More than 7	18	43	38	18
Consumption frequency	Every day	2	4	10	0
	Every 3 days	8	14	21	4
	Once a week	15	14	15	23
	Every 15 days	37	22	29	21
	Once a month	38	46	25	52

**Table 2 foods-11-03549-t002:** Constructs and corresponding items evaluated in the survey.

Construct	Used Item	Factor Loading
Food Involvement [11]	Drinking mezcal is an important part of my social life.	0.83
	Drinking mezcal is an important part of my life.	0.50
	Drinking mezcal is a continuous source of joy for me.	0.83
	Decisions about what mezcal to drink are very important for me.	0.78
	I just love good mezcal.	0.84
Objective knowledge own elaboration	Appendix A	
Novelty [10]	I love trying mezcal from foreign regions.	0.89
	I like to try mezcal that I have never tasted before.	0.88
Price criterion [10]	I always check prices, even on small items.	0.64
	I notice when I buy mezcal it regularly changes in price.	0.79
Price–quality relationship [10]	I always try to get the best quality of mezcal for the best price.	0.84
	I compare prices between product variants in order to get the best mezcal for the money.	0.84
	It is important for me to know that I get quality for all my money.	0.88
Product information [10]	To me, product information is of major importance. I want to know what mezcal contains.	0.50
	I compare labels to select the best mezcal.	0.99
	I compare product information labels to decide which mezcal to buy.	0.80
Self-fulfilment [10]	Being praised for drinking mezcal adds a lot to my self-esteem.	0.53
	Drinking mezcal is, to me, a matter of touching, smelling, tasting and seeing, all the senses are involved. It is a very exciting sensation.	0.50
	I am an excellent mezcal taster.	0.95
Social events [10]	Going out for mezcal is a regular part of our drinking habits.	0.84
	We often get together with friends to enjoy casual drinks like mezcal.	0.84
	I do not consider it a luxury to go out with my family to have mezcal in a restaurant.	0.83
Social relationships [10]	I find that drinking mezcal with friends is an important part of my social life.	0.69
	When I serve mezcal to friends, the most important thing is that we are together.	0.69
Subjective knowledge [19]	When it comes to mezcal, I know a lot.	0.80
	I know pretty much about mezcal.	0.56
	Among my circle of friends, I’m one of the “experts” on mezcal.	0.70
	Compared to most people, I know less about mezcal.	0.64
	I do not feel very knowledgeable about mezcal.	0.69

All items with factor loadings lower than 0.5 were removed from the original scale. Numbers in bold indicate significant values.

**Table 3 foods-11-03549-t003:** ANOVA of constructs among the cities.

Construct	Guadalajara	Mexico City	Oaxaca	Puebla
Involvement	3.33	3.47	4.00	3.44
Objective knowledge	8.60 b	8.64 b	11.34 a	8.25 b
Novelty	5.83	6.26	6.13	6.13
Price criterion	5.00	4.61	5.00	4.75
Price–quality relationship	4.60 a	3.89 bc	3.78 c	4.56 ab
Product information	4.76	4.36	4.60	4.26
Self-fulfillment	3.55 b	3.80 ab	4.14 a	3.64 ab
Social events	4.15 ab	4.07 ab	4.60 a	3.87 b
Social relationships	4.34	4.24	4.48	4.34
Subjective knowledge	3.46 b	3.73 ab	4.26 a	3.79 ab

Different letters in the same row indicate significant differences among the cities (*p* < 0.05).

**Table 4 foods-11-03549-t004:** Proportions of selected words per city to describe mezcal.

Descriptor	Guadalajara	Mexico City	Oaxaca	Puebla
Agave	0.49 b	0.68 ab	0.77 a	0.61 ab
Alcohol	0.38	0.44	0.52	0.43
Bottle	0.31	0.41	0.42	0.32
Blue agave	0.00 b	0.10 a	0.10 a	0.08 a
Burning	0.31	0.21	0.16	0.27
Cheap	0.08	0.14	0.05	0.15
Commercial	0.04	0.13	0.13	0.07
Culture	0.75	0.77	0.75	0.74
Drink	0.48 a	0.27 b	0.34 ab	0.26 b
Drunk	0.26 a	0.08 b	0.23 a	0.21 ab
Drunkenness	0.27	0.13	0.25	0.26
Food	0.17 b	0.44 a	0.39 a	0.39 a
Handcraft	0.83	0.87	0.93	0.89
Hangover	0.15	0.10	0.09	0.17
Industrialized	0.03	0.02	0.02	0.00
Industry	0.04	0.04	0.07	0.02
Maguey	0.47 b	0.58 b	0.81 a	0.52 b
Mariachi	0.09	0.13	0.08	0.12
Mexican	0.81	0.77	0.70	0.75
Nice	0.37 b	0.62 a	0.52 ab	0.55 ab
Orange	0.52 ab	0.56 ab	0.38 b	0.61 a
Paloma cocktail	0.02	0.03	0.01	0.02
Party	0.38 ab	0.32 b	0.57 a	0.37 ab
Smoked	0.67	0.68	0.70	0.63
Soft	0.15 ab	0.20 ab	0.30 a	0.11 b
State	0.41	0.39	0.47	0.35
Strong	0.49	0.50	0.50	0.44
Tradition	0.55 b	0.65 ab	0.75 a	0.78 a
Worm	0.59	0.75	0.70	0.76

Different letters in the same row mean significant differences among the cities according to the chi-square test (*p* < 0.05) of the K proportion test with the Marascuilo procedure.

**Table 5 foods-11-03549-t005:** Segment identification based on assessed constructs for mezcal.

S	Involvement	Subjective Knowledge	Objective Knowledge	Quality Price	Novelty	Product Info.	Price Criterion	Social Events	Social Relations	Self Fulfilment
1	5.65 a	5.35 a	10.84 a	4.53 b	6.79 a	5.3 a	5.50a	5.92 a	5.98 a	5.18 a
2	2.58 c	2.68 c	6.56 b	6.17 a	6.57 ab	5.4 a	6.04 a	3.55 c	4.01 b	3.37 b
3	3.25 b	3.85 b	9.89 a	3.82 c	6.40 b	4.3 b	4.50 b	4.18 b	4.39 b	3.67 b
4	2.01 d	2.48 c	7.61 b	3.33 c	4.51 c	3.2 c	3.77 c	2.34 d	2.47 c	2.46 c

S, Segment; different letters in the same column mean significant differences among the segments.

**Table 6 foods-11-03549-t006:** Proportions of sociodemographic characteristics, consumption habits, and product experience of mezcal segments (*N* = 400).

Categories	Cluster 1 *n* = 115	Cluster 2 *n* = 51	Cluster 3 *n* = 140	Cluster 4 *n* = 94
Gender				
Male	65.2	53.0	61.0	56.4
Female	34.8	47.0	39.0	43.6
City				
Guadalajara	22.6	41.2	22.9	22.3
Mexico City	26.1	27.4	22.9	25.5
Oaxaca	33.0 a	5.9 b	25.0 a	25.6 a
Puebla	18.3	25.5	29.2	26.6
Age (years)				
18–24	11.3	17.6	15.7	19.1
25–34	38.3	43.1	42.9	38.3
35–44	29.5	21.6	31.4	25.6
45–54	17.4	11.8	7.9	11.7
55 or more	3.5	5.9	2.1	5.3
Education				
Undergraduate	21.7 b	45.1 a	27.9 ab	35.1 ab
University degree	78.3 a	54.9 b	72.1 ab	64.9 ab
Experience with the product				
Less than a year	1.7 b	27.5 a	6.4 b	23.4 a
From 1 to 3 years	10.4 b	21.6 ab	25.0 a	26.6 a
From 3 to 5 years	15.7 a	25.5 a	22.9 a	19.1 a
From 5 to 7 years	24.4 a	7.8 b	20.7 ab	11.8 ab
More than 7 years	47.8 a	17.6 b	25.0 b	19.1 b
Consumption frequency				
Every day	9.6 a	0.0 b	2.9 ab	1.1 b
Every third day	29.6 a	4.0 b	5.0 b	4.3 b
Once a week	27.8 a	7.8 b	15.0 ab	20.2 ab
Every 15 days	22.6 ab	25.5 ab	36.4 a	10.6 b
Once a month	10.4 c	62.7 ab	40.7 b	63.8 a

Different letters in the same row mean significant differences among the clusters according to the chi-square test (*p* < 0.05) of the K proportion test with the Marascuilo procedure.

**Table 7 foods-11-03549-t007:** Proportions of selected words per segments to describe mezcal.

Descriptor	Segment 1	Segment 2	Segment 3	Segment 4
Agave	0.74 b	0.45 a	0.67 ab	0.55 a
Alcohol	0.40	0.43	0.42	0.52
Bottle	0.38	0.25	0.37	0.39
Blue agave	0.04	0.07	0.06	0.10
Burning	0.13 a	0.19 ab	0.28 b	0.30 b
Cheap	0.07 a	0.09 ab	0.05 a	0.22 b
Commercial	0.09	0.05	0.09	0.10
Culture	0.84 b	0.60 a	0.76 ab	0.70 ab
Drink	0.34	0.37	0.31	0.28
Drunk	0.16	0.15	0.17	0.27
Drunkenness	0.21	0.11	0.22	0.29
Food	0.52 c	0.13 a	0.33 b	0.26 ab
Handcraft	0.87	0.92	0.82	0.86
Hangover	0.07 a	0.07 a	0.13 ab	0.21 b
Industrialized	0.01	0.00	0.01	0.03
Industry	0.04	0.02	0.05	0.04
Maguey	0.76 b	0.37 a	0.55 a	0.56 a
Mariachi	0.03 a	0.11 ab	0.12 ab	0.16 b
Mexican	0.78	0.80	0.75	0.69
Nice	0.64 c	0.41 ab	0.58 bc	0.30 a
Orange	0.49	0.52	0.49	0.51
Paloma cocktail	0.00	0.02	0.02	0.03
Party	0.47	0.27	0.40	0.41
Smoked	0.74	0.52	0.71	0.59
Soft	0.25	0.21	0.16	0.13
State	0.47	0.27	0.42	0.37
Strong	0.47	0.35	0.50	0.53
Tradition	0.81 c	0.45 a	0.65 ab	0.69 bc
Worm	0.65	0.56	0.75	0.75

Different letters in the same row mean significant differences among clusters according to the chi-square test (*p* < 0.05) of the K proportion test with the Marascuilo procedure.

## Data Availability

All data are presented in this manuscript.

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
