# Peer review of "Conceptualizing a Product with the Food-Related Lifestyle Instrument"

_foods, 2022, doi:10.3390/foods11223549_

Round 1

Reviewer 1 Report

The article deals with an interesting topic Mezcal conceptualization according to the Food Related Lifestyle instrument. But it has a few major limitations.

 (1) Novelty. What is novel to investigate conceptual characterization of food? Lots of similar researches exist. The authors should outline how the main findings are different with previous studies. Please compare your findings with more other studies, especially in the discussion section.

(2) Logic is confused in the introduction section. There are too many nouns, and authors needs to specify the difference and connection of these nouns, such as food perception, conceptualization, cognitive, and lifestyle.

(3) What is the purpose of this study? Why study the impact of geographical location. The impacts of geographical location was not significant accoding to the data in Table 3 and 4.

(4) Four hundred consumers from four different Mexican cities participated in this study. How were the participants selected?

Author Response

First, authors of this work would like to thank the reviewer for the insights and comments for this paper. The anwers to all of your comments can be seen in the attached file.

English has been edited by a professional in order to improve the quality as suggested

Reviewer 2 Report

The title is a little confusing; maybe it could be changed to something simpler and clearer.

e.g., “Conceptualizing Mezcal consumption according to the Food Related Lifestyle instrument” or any other suitable title that better represents the current study.

Introduction

Background information about Mezcal is needed. This journal publishes and targets worldwide readers. Thus, it is necessary to explain what Mezcal is, its history, and why it is important to the people of Mexico and their culture. Why does the international audience need to know about Mezcal and the outcome of this study? Some statistical data and current market trends will assist readers in better grasping what is going on in the industry.

What exactly is Mezcal? Why is it important to study Mezcal and the different segments of its market? How many units of this beverage are being sold each day? Is there more than one type of Mezcal available?

Literature Review: There is no proper literature review section in this paper. There is a need to discuss the Food Related Lifestyle instrument, the theory, and various studies on this subject matter—the justification for why this instrument has been selected.

 Methods and Results: Adequate

Discussion:  The ramifications of the research may need to be discussed or given their own section. How can the result benefit the industry make use of it, taking into consideration the specified segments?

Author Response

The authors of this work are greatful for your input in this paper.

Manuscript was run by a proffesional for English improvement as suggested.
